



# Prediction of Secondary Organic Aerosol from the Multiphase Reaction of Gasoline Vapor by Using Volatility–Reactivity Base Lumping

Sanghee Han[1], Myoseon Jang[1]

[1]Department of Environmental Engineering Science, University of Florida, Gainesville, Florida, USA

*Correspondence to:* Myoseon Jang (mjang@ufl.edu)

**Abstract.** The secondary organic aerosol (SOA) formation from photooxidation of gasoline vapor was simulated by using the UNIfied Partitioning Aerosol phase Reaction (UNIPAR) model, which predicted SOA growth via multiphase reactions of hydrocarbons. The Carbon Bond 6 (CB6r3) mechanism was incorporated with the SOA model to estimate the hydrocarbon consumption and the concentration of radicals (i.e., $RO_2$ and $HO_2$), which were closely related to atmospheric aging of gas products. Oxygenated products were lumped according to their volatility and reactivity and linked to stoichiometric coefficients and their physicochemical parameters, which were dynamically constructed at different $NO_x$ levels and degrees of gas aging. To assess the gasoline SOA potential in ambient air, model parameters were corrected for gas–wall partitioning (GWP), which was predicted by a qualitative structure activity relationship for explicit products. The simulated gasoline SOA mass was evaluated against observed data obtained in the UF-APHOR chamber under ambient sunlight. The influence of environmental conditions on gasoline SOA was characterized under varying $NO_x$ levels, aerosol acidity, humidity, temperature, and concentrations of aqueous salts and gasoline vapor. Both the measured and simulated gasoline SOA formation was sensitive to seeded conditions (acidity and hygroscopicity) and $NO_x$ levels. A considerable difference in SOA mass appeared before and after efflorescence relative humidity in the presence of salted aqueous solution. SOA growth in the presence of aqueous reactions was more impacted by temperature than that in absence of seed. The impact of GWP on SOA formation was generally significant, and it appeared to be higher in the absence of wet salts. We conclude that the SOA model in the corpus with both heterogeneous reactions and the model parameters corrected for GWP is essential to accurately predict SOA mass in ambient air.

## 1 Introduction

The atmospheric oxidation of hydrocarbons (HCs) produces ozone in the troposphere through a photochemical cycle of nitrogen oxides. This ozone can increase the risks of respiratory disease (Jerrett et al., 2009), cardiovascular disease, and premature death (Turner et al., 2016). Additionally, the atmospheric process of HCs can produce semivolatile oxygenated products that can form secondary organic aerosol (SOA) through either gas–particle partitioning or aerosol phase reactions. SOA is a major contributor to the fine particular matter ($PM_{2.5}$) in ambient air (Jimenez et al., 2009), with significant effects on climate (Heald et al., 2008) and human health (Cohen et al., 2017;Pöschl, 2005). Thus, it is critical to understand and predict the amount of SOA produced from the atmospheric oxidation of various HCs to effectively improve air quality and human health.

The air quality models, integrated with the partitioning based SOA module using two (Odum et al., 1996) or several surrogate species (i.e., Volatility Basis Set (Donahue et al., 2006)), tend to underpredict SOA budget in the urban ambient air (Volkamer et al., 2006;Dzepina et al., 2011;Ensberg et al., 2014;Hayes et al., 2015;Appel et al., 2017;Woody et al., 2016). Much effort has been given to reduce the model–measurement discrepancies by adding missing SOA precursors (McDonald et al., 2018), including heterogeneous reactions (Carlton et al., 2010), and correcting the SOA model parameters by considering gas–wall



partitioning (GWP) bias(Cappa et al., 2016;Baker et al., 2015;Hayes et al., 2015). For example, high SOA yields (>0.1) were reported from several individual intermediate volatility compounds in consumer product mixtures (Li et al., 2018). Thus, the

extension of the SOA model to additional precursors emitted from the commercial usage of chemicals was attempted to close the model-measurement gap (Qin et al., 2021;Shah et al., 2019;McDonald et al., 2018).

The oligomers in SOA formed from photooxidation of precursor HCs in chambers and the ambient air has been identified as 25-70% of the SOA mass (Kalberer et al., 2004;Tolocka et al., 2004;Gross et al., 2006;Kalberer et al., 2006;Hallquist et al., 2009). Thus, there have been numerous implications of in-particle chemistry in SOA formation for the model studies (Carlton

and Turpin, 2013;Carlton et al., 2010;Pye et al., 2017). For example, the formation of oligomers was considered in the SOA module of Community Multiscale Air Quality (CMAQ) as a first order reaction of condensed organic species, resulting in the improvement of spatial and temporal trends of SOA mass in particular for biogenic SOA (Carlton et al., 2010). To introduce the role of aerosol water contents in SOA formation, Jathar et al. (2016) examined the water uptake to the organic phase in SOA model and assess its influence on SOA formation (Jathar et al., 2016). Moreover, Pye et al. (2017) evaluate the importance

of aerosol-water-organic interactions in the CMAQ model accounting for the uptake of water onto the hydrophilic organics (Pye et al., 2017). Despite such efforts, the performance of SOA formation in representing spatial and seasonal variation in ambient aerosol tends to underestimate total aerosol mass in the southern and western US (Appel et al., 2021).

Inaccuracy of SOA predictions can be originated from the integration of the SOA model with the model parameters originated from biased chamber data. Laboratory studies have shown that the deposition of organic vapor onto the reactor wall can cause

the negative bias in SOA model parameters (Matsunaga and Ziemann, 2010;Zhang et al., 2014;Yeh and Ziemann, 2015;Krechmer et al., 2016;Huang et al., 2018). Thus, the GWP of semivolatile organic compounds was semiempirically characterized for various organic species. Many model studies reported that the significance of GWP varied with precursor HCs, gas oxidation process (different $NO_x$ levels or oxidants), chamber dimension and size, seed conditions, and meteorological conditions (temperature and humidity) (Zhang et al., 2014;Krechmer et al., 2020;Brune, 2019;Huang et al.,

2018). To simulate SOA formation accurately, the integration of the GWP model to the SOA model is inevitable in chamber studies. However, the application of individually determined GWP bias in the air quality model is challenging due to the surrogate species in air quality models.

In this study, the SOA formation from photooxidation of gasoline vapor was simulated with the UNIfied Partitioning Aerosol-phase Reaction (UNIPAR) model, which predicts SOA formation via multiphase reactions. The important feature of the

UNIPAR model is to simulate SOA formation via aqueous phase reactions of organic species in the absence of GWP. The UNIPAR model streamlines gas oxidation mechanisms, multiphase partitioning (gas, organic phase, and inorganic salted solution), and aerosol-phase reactions in both organic and inorganic phases (Beardsley and Jang, 2016;Im et al., 2014;Zhou et al., 2019;Yu et al., 2021). The model parameters and equations in the model have been demonstrated for various SOA produced from aromatic HCs (Im et al., 2014;Zhou et al., 2019), terpenes (Yu et al., 2021), and isoprene (Beardsley and Jang, 2016). In

order to predict SOA mass in ambient air, the model parameters were updated by using the GWP model that employed a quantitative structure activity relationship (QSAR) approach (Han and Jang, 2020). The Carbon Bond 6 (CB6r3) (Yarwood et al., 2010) mechanism was integrated with UNIPAR to obtain the consumption of HCs and the concentration of radicals ($RO_2$ and $HO_2$) that processed atmospheric aging. The simulated SOA mass was compared to chamber generated SOA data under the University of Florida Atmospheric PHotochemical Outdoor Reactor (UF-APHOR). The UNIPAR prediction was

also compared to the prediction from the AERO7 module in CMAQ model (CMAQ-AE7)(Appel et al., 2021). The sensitivity of gasoline SOA formation to various environmental conditions, such as temperature, relative humidity (RH), seed conditions, and the concentration of HC, was investigated.



## 2 Chamber experiment

Gasoline SOA was generated from the photooxidation of US commercial gasoline vapor (octane numbers of 87) under ambient
sunlight in the UF-APHOR outdoor chamber located on the rooftop of Black Hall (29.64˚, -82.34˚) at the University of Florida,
Gainesville, Florida. Based on the gas chromatography-flam ionization detector (GC–FID, HP-5890/Agilent Technologies
7820A) analysis of injected gasoline vapor, 30% of carbons in the gasoline were from single-ring aromatic HCs (Fig. S1). The
gasoline vapor, $NO_x$, and inorganic seed aerosols were injected into the chamber before sunrise, and experiments were
conducted for 10 hours after sunrise. The $NO_x$ level was classified into high $NO_x$ ($HC/NO_x < 5.5$ ppbC/ppb) and low $NO_x$ level
($HC/NO_x > 5.5$ ppbC/ppb) based on the initial concentration of HC and $NO_x$. Four different seed conditions (non-seeded (NS);
sulfuric acid (SA); wet ammonium sulfate (wAS); dry ammonium sulfate (dAS)) were applied to evaluate the seed effects on
gasoline SOA. The chamber conditions for conducted experiments were summarized in Table 1.

The concentration of HCs and $CCl_4$ were monitored using a GC–FID. The measured HC and $CCl_4$ concentration from GC–
FID determined HC consumption and dilution in the chamber during the experiment, respectively. The concentrations of $O_3$
and $NO_x$ were monitored with a photometric ozone analyzer (Teledyne, model 400E) and a chemiluminescence $NO/NO_2$
analyzer (Teledyne, model 200E), respectively. The measurement of inorganic ion ($SO_4^{2-}$ and $NH_4^+$) and organic carbon (OC)
concentrations of aerosol were conducted with in situ monitoring by the Particle-Into-Liquid-Sampler (Applikon, ADI 2081)
coupled with Ion Chromatography (Metrohm, 761Compact IC) (PILS–IC) and an OC/EC carbon aerosol analyzer (Sunset
Laboratory, Model 4), respectively. The particle volume concentration was monitored with a Scanning Mobility Particle Sizer
(SMPS, TSI, Model 3080) integrated with a condensation nuclei counter (TSI, Model 3025A and Model 3022). The
composition ($SO_4^{2-}$, $NO_3^-$, $NH_4^+$, and organic) of aerosol was also monitored using an Aerosol Chemical Speciation Monitor
(ACSM, Aerodyne Research Inc., USA) to compare with the data obtained from OC and PILS–IC for the accurate
measurement. The meteorological factors (RH, temperature, and ultraviolet radiation) were measured in the UF-APHOR and
applied to the simulation. Aerosol acidity (mol/L of aerosol) was monitored by colorimetry integrated in the reflectance UV–
visible spectrometer (Li and Jang, 2012;Jang et al., 2020).

## 3 Model descriptions

The UNIPAR model was coupled with CB6r3 gas mechanism to simulate SOA formation from photooxidation of HCs
(UNIPAR-CB6r3). The overall structure of the UNIPAR-CB6r3 model is illustrated in Fig. 1. The main feature of the model
is the SOA simulation based on volatility and reactivity of organic products by using lumping species that are constructed from
explicit gas products. In UNIPAR-CB6r3, the model parameters are corrected by the GWP artifacts, and they are also
universalized for 10 aromatic HCs to predict SOA formation from gasoline vapor. The UNIPAR-CB6r3 model was simulated
in the Dynamically Simple Model of Atmospheric Chemical Complexity (DSMACC) (Emmerson and Evans, 2009) integrated
with the Kinetic PreProcessor (KPP) (Damian et al., 2002).

The CB6r3 mechanism simulates the atmospheric oxidation of anthropogenic precursors in the gas phase and yields the
consumption of HCs (ΔHC) and concentration of $RO_2$ ([$RO_2$]) and $HO_2$ ([$HO_2$]). In the model, the predetermined polynomial
equations, derived by using explicit gas mechanisms (Master Chemical Mechanism, MCM v3.3.1) (Jenkin et al., 2012),
estimate the stoichiometric coefficients ($\alpha_i$) of the lumping species ($i$) of gas phase oxygenated products. The quantity of $\alpha_i$
array, which comprises 51 lumping species according to their volatility and reactivity in the aerosol phase, are dynamic as a
function of a $HC/NO_x$ level and degree of aging, which is calculated with ΔHC, [$RO_2$], and [$HO_2$] from CB6r3. These lumping
species are then used to generate SOA mass ($OM_T$) via gas–particle partitioning ($OM_P$) and heterogeneous reactions ($OM_{AR}$)
in both organic and inorganic phases. The SOA formation via aqueous phase reactions of organic species was simulated under





the assumption of the liquid-liquid phase separation (LLPS) between organic and inorganic phase. The details of the model description are shown in the following sections.

### 3.1 Lumped Organic Species

The gas oxidation products are categorized into 51 lumping species based on their volatility and reactivity in aerosol phase to predict SOA formation through the multiphase reactions. These lumping species are linked to a mass-based stoichiometric coefficient ($\alpha_i$) to produce the concentrations of each lumping species. In order to process the multiphase thermodynamic equilibrium of lumping species ($i$), the physicochemical properties (vapor pressure ($p_{L,i}^{\circ}$), molecular weight ($MW_i$), oxygen to carbon ratio ($O:C_i$), and hydrocarbon bond ($HB_i$)) are determined based on the group contribution (Stein and Brown, 1994).

In the UNIPAR model, the volatilities of each oxidation product are classified into eight levels of the vapor pressure ($P_{L,i}^{\circ}$) (1–8: $10^{-8}$, $10^{-6}$, $10^{-5}$, $10^{-4}$, $10^{-3}$, $10^{-2}$, $10^{-1}$, and 1 mmHg) and six levels based on the aerosol phase reactivity scale ($R_i$): very fast (VF), fast (F), medium (M), slow (S), partitioning only (P), and multi-alcohol (MA) and three additional reactive species (glyoxal, methylglyoxal, and epoxydiols).

In UNIPAR-CB6r3, the UNIPAR model is coupled with the CB6r3 mechanism (Yarwood et al., 2010) that is used in air quality models. The CB6r3 mechanism has been frequently used to predict the ozone formation by using a unique lumping species (Yarwood et al., 2005;Yarwood et al., 2010) but is limited to consider the physicochemical properties of oxygenated products, which are necessary to process SOA formation. In this study, ΔHC, [RO₂], and [HO₂] are obtained from CB6r3 and incorporated with predetermined polynomial equations to determine the $\alpha_i$ array as a function of HC/NOₓ ratios and an aging scale (Fig. 1). In CB6r3, the reaction of toluene or benzene with an OH radical is explicitly expressed. In order to obtain ΔHC

from each aromatic HC, the reactions of other aromatic HCs (i.e., ethylbenzene, propylbenzene, 3 xylene isomers, and 3 trimethylbenzene isomers) with an OH radical were explicitly treated by using individual reaction rate constants (Table S1). The physicochemical parameter arrays ($MW_i$, $O:C_i$, and $HB_i$) of lumping species are also dynamically built with predetermined polynomial equations. These equations are mathematically derived by using explicit products predicted from MCM v3.3.1. The gas concentration ($C_{g,i}$) of lumping species, $i$, is estimated by multiplying ΔHC by dynamically predicted

$\alpha_i$ (Zhou et al., 2019).

### 3.2 Multiphase Partitioning

The gas products produced from the atmospheric oxidation of precursor HCs partition onto both organic and inorganic phase ($C_{or,i}$ and $C_{in,i}$, respectively). $C_{or,i}$ and $C_{in,i}$ are incorporated into in-particle chemistry to form $OM_{AR}$ (Fig. 1). In this model, the gas–particle partitioning processes are assumed as an equilibrium partitioning process based on the absorptive partitioning

theory (Pankow, 1994), which assumes that the gas–particle partitioning instantaneously reaches equilibrium to distribute the gas products into the gas, organic and inorganic phases.

The partitioning coefficient of $i$ into the organic phase ($K_{or,i}$) and inorganic phase ($K_{in,i}$) are determined by the traditional absorptive partitioning theory (Pankow, 1994) as follows:

$$K_{or,i} = \frac{7.501RT}{10^9 MW_{or}\gamma_{or,i}p_{L,i}^{\circ}}, \tag{1}$$

$$K_{in,i} = \frac{7.501RT}{10^9 MW_{in}\gamma_{in,i}p_{L,i}^{\circ}}, \tag{2}$$

where $MW_{or}$ (g mol⁻¹) is the molecular weight of $OM_T$, R (8.314 J mol⁻¹ K⁻¹) is the ideal gas constant, and T (K) is the temperature. $\gamma_{or,i}$ is the activity coefficient of $i$ in organic phase. $MW_{in}$ is the molecular weight of $M_{in}$, and $\gamma_{in,i}$ is the activity coefficient of $i$ in inorganic phase. $\gamma_{or,i}$ is assumed as unity, while $\gamma_{in,i}$ is semiempirically estimated with a polynomial





equation, determined by fitting the $\gamma_{in,i}$ estimated by the Aerosol Inorganic–Organic Mixtures Functional groups Activity

Coefficient (AIOMFAC) (Zuend et al., 2011):

$$\gamma_{in,i} = e^{0.035MW_i - 2.704\ln(O:C_i) - 1.121HB_i - 0.33FS - 0.022(RH)},$$  (3)

where RH is relative humidity (%), and FS (fractional sulfate, FS = [SO$_4^{2-}$]/([SO$_4^{2-}$]+[NH$_4^+$])) is the concentration ratio of total

sulfate to the sum of total sulfate and ammonium ions in aerosol (Zhou et al., 2019). FS is 1 in SA and 0.33 in AS, indicating

the aerosol acidity. To simulate the gasoline SOA, unified matrix of $MW_i$, $O:C_i$, and $HB_i$ for 51 lumping species were applied

to the SOA prediction from 10 different aromatic HCs, which are the compositions of gasoline vapor.

### 3.3 OM$_{AR}$: SOA growth via aerosol phase reactions

OM$_{AR}$ is produced via oligomerization in both organic and inorganic phases, as described in previous studies (Im et al.,

2014;Zhou et al., 2019). OM$_{AR}$ is estimated as a second order reaction product from condensed organics based on the

assumption of a self-dimerization reaction of organic compounds in media (Odian, 2004):

$$\frac{dC'_{or,i}}{dt} = -k_{o,i}C'^{2}_{or,i},$$  (4)

$$\frac{dC'_{in,i}}{dt} = -k_{AC,i}C'^{2}_{in,i},$$  (5)

where $C'_{or,i}$ and $C'_{in,i}$ are the concentration of $i$ in the organic and inorganic aerosol phase (mol L$^{-1}$), respectively. The reaction

rate constant in the aqueous phase ($k_{AC,i}$) and organic phase ($k_{o,i}$) are determined:

$$k_{AC,i} = 10^{0.22pK_{BH_i^+} + 0.6X + 0.846R_i + \log(a_w[H^+]) - 2.2},$$  (6)

$$k_{o,i} = 10^{0.25pK_{BH_i^+} + 0.95R_i + 5.2\left(1 - \frac{1}{1+e^{-0.2(MW_{or} - 270)}}\right) + \frac{2.7}{1+e^{-11(O:C - 0.85)}} - 10.07},$$  (7)

where $k_{AC,i}$ is semiempirically defined as a function of $R_i$, the protonation equilibrium constant ($pK_{BH_i^+}$), excess acidity (X)

(Cox and Yates, 1979;Jang et al., 2006), water activity ($a_w$), and the proton concentration [H$^+$] (Im et al., 2014;Zhou et al.,

2019). $k_{o,i}$ is determined without X, $a_w$, and [H$^+$] terms because $a_w$, [H$^+$], and X converged to zero in the absence of wet

inorganic seed.

SOA formation in the aqueous phase may not be affected by viscosity because of abundant water molecules, while SOA

formation in organic phase can be impacted by viscosity. In the traditional SOA models, it often assumes that the equilibrium

in gas–particle partitioning is rapidly achieved. However, studies have shown the relatively long characteristic time of less

volatile organic species onto the semisolid particle to reach equilibrium (Shiraiwa and Seinfeld, 2012;Shiraiwa et al., 2011).

In general, the higher viscosity appears with an organic compound with the higher MW (Koop et al., 2011). Although the high

O:C ratio (higher polarity) can increase viscosity, polar aerosol can absorb water in some degree at high humidity increasing

fluidity. Studies showed that viscosity can also influence chemical reaction rates (De Schrijver and Smets, 1966;Reid et al.,

2018). In the current knowledge, it is difficult to distinguish the impact of viscosity on the partitioning process from that on

reaction rates. In this study, the impact of aerosol viscosity on SOA formation was related to heterogeneous reactions in the

organic phase by controlling their reaction rate constant. The sigmoidal functions for $MW_{or}$ and $O:C$ are involved in $k_{o,i}$ to

consider the viscosity of organic aerosol.

### 3.4 OM$_P$: SOA formation via partitioning

It is assumed that gas–organic partitioning is governed by Raoult's law in that the saturation vapor pressure of the species is

dependent on the mole fraction of the species in the solution. To consider the subtracted mass in total concentration ($C_{T,i} =$

$C_{g,i} + C_{or,i} + C_{in,i}$) of $i$ by oligomerization, OM$_P$ is recalculated after OM$_{AR}$ integration with the partitioning model (Schell et





al., 2001) which is reconstructed by including $OM_{AR}$ (Cao and Jang, 2010). $OM_P$ is estimated from $C_{T,i}$ using a mass balance

equation and solved by the Newton Raphson method (Press et al., 1992):

$$OM_P = \sum_i \left[ C_{T,i} - OM_{AR,i} - C_{g,i}^* \frac{\left( \frac{C_{or,i}}{MW_i} \right)}{\sum_i \left( \frac{C_{or,i}}{MW_i} + \frac{OM_{AR,i}}{MW_{oli,i}} \right) + OM_0} \right] , \tag{8}$$

where $C_{g,i}^* \left( = \frac{1}{K_{or,i}} \right)$ is the effective saturation concentration of $i$, $OM_0$ (mol m$^{-3}$) is the pre-existing OM concentration, and

$MW_{oli,i}$ (g mol$^{-1}$) is the molecular weight of the dimer.

**3.5 Model parameters in the absence of GWP bias**

The model parameters inherited from chamber-generated SOA mass can be biased and increase inaccuracy in prediction of

SOA mass in ambient air. In this study, the UNIPAR-CB6r3 simulation was performed with the model parameters that were

not biased by GWP. Regardless of the absence or the presence of GWP, the prediction of $OM_P$ is approached by a fundamental

partitioning theory. The modification was mainly made for the model parameters associated with aerosol phase reaction rate

constants of lumping species (i.e., $k_{o,i}$). Consequently, the change in $OM_{AR}$, predicted with the new model parameters,

modulates $OM_P$ (Fig. 1).

In order to establish $k_{o,i}$ in the absence of GWP on the model, UNIPAR was integrated with explicit gas mechanisms (MCM

v3.3.1) and the GWP model (UNIPAR-GWP, in Sect. S2) (Han and Jang, 2020). Multiphase HC reactions, including gas phase

oxidation, gas–particle partitioning, aerosol phase reactions in organic phase and inorganic aqueous phase, and GWP, were

kinetically treated in the model (Fig. S2). In UNIPAR-GWP, the GWP processes were predicted with GWP model parameters

(organic vapor-wall partitioning and its accommodation coefficient) and the chamber specific characteristics (chamber

dimension and the organic matter concentration on the wall). The QSAR approach that employed organic physicochemical

parameters facilitated the derivation of the semiempirical polynomial model to predict GWP model parameters, as described

in the previous study (Han and Jang, 2020). The $k_{o,i}$ values were amended to predict aromatic SOA by using UNIPAR-GWP

against chamber-generated data. The amended $k_{o,i}$ was then applied to the UNIPAR-CB6r3 model to predict the gasoline SOA

formation potential in this study.

**4 Results and Discussions**

**4.1 Aromatic SOA simulation with UNIPAR-CB6r3**

The gasoline fuel is the mixture of various HCs including aromatics, alkenes, alkanes, and ethanol. It has been known that the

photooxidation of aromatic HCs in the presence of NO$_x$ mainly attributes to gasoline SOA (Gentner et al., 2017). Thus, the

feasibility of UNIPAR-CB6r3 was evaluated for the prediction of the SOA formation from photooxidation of aromatic HCs in

gasoline against chamber-generated SOA data (Table S2). In Fig. 2, the simulation was plotted against chamber-generated

SOA mass produced from the photooxidation of 10 different aromatic HCs under various experimental conditions (Table S2).

Overall, the predicted SOA mass with UNIPAR-CB6r3 in the presence of GWP agreed well with the observed SOA mass

(slope = 1.04 and R$^2$ = 0.89). Figure S3 illustrates the simulated (solid line) and observed (symbol) concentration of NO, NO$_2$,

O$_3$ and HC over the course of the experiment. Figure S4 shows a time profile of simulated $OM_T$ (solid line), simulated $OM_P$

(dotted line) and observed SOA data (symbol) over the course of the chamber experiment. A significant contribution of $OM_{AR}$

to $OM_T$ suggests an importance of aerosol phase heterogeneous reactions in SOA formation. The oligomers in SOA formed

from photooxidation of precursor HCs in chambers and the ambient air has been identified as a considerable fraction in the

SOA mass (Kalberer et al., 2004;Tolocka et al., 2004;Gross et al., 2006;Kalberer et al., 2006;Hallquist et al., 2009).



To characterize the impact of GWP on aromatic SOA formation, the aromatic SOA mass ($C_{SOA}$, $\mu$g m$^{-3}$) in the absence of GWP bias was plotted versus the aromatic SOA mass ($C_{SOA,wall}$, $\mu$g m$^{-3}$) predicted in the presence of the GWP (Fig. S5). The SOA formation was simulated at (a) high NO$_x$ level (HC/NO$_x$= 3 ppbC/ppb) and (b) low NO$_x$ level (HC/NO$_x$= 10 ppbC/ppb) at the given reference conditions (298K, 60% RH) under the specific sunlight intensity measured on 06/19/2015 (Fig. S6). The more deviated plot from the 1:1 line indicates the larger impact of GWP on SOA formation. Impact of GWP on SOA formation differs with oxidation product distributions according to volatility and reactivity. The HC with the higher impact of OM$_{AR}$ on OM$_T$ is less influenced by GWP. For example, benzene SOA, which is mainly attributed to OM$_{AR}$, was little influenced by GWP. The impact of GWP on SOA formation in the presence of inorganic seed (wAHS in red and wAS in blue) was significantly less than that without the wet inorganic seed (black). In the presence of wet seed, plots are much less deviated from the one-to-one line as seen in Fig. S5 in comparison to no-seeded SOA. This tendency agreed with the previously reported results (Krechmer et al., 2020;Zhang et al., 2014). The SOA formation pathway via aqueous reactions of organic products show little sensitivity to GWP.

### 4.2 Gasoline SOA simulation with UNIPAR-CB6r3

Figure 3 shows the time profiles of simulated OM$_T$ (solid line), OM$_{AR}$ (dotted line), and observed organic matter (symbols) under the experimental conditions summarized in Table 1. Overall, the gasoline SOA mass predicted by UNIPAR-CB6r3 in the presence of wall agrees with SOA mass generated in UF-APHOR under varying NO$_x$ levels and seed conditions. Similar to aromatic SOA (Fig. S4), gasoline SOA formation was dominated by OM$_{AR}$. SOA mass at the low NO$_x$ level is higher than that at the high NO$_x$ level (Fig. 3(a) vs. 3(b)), causing higher contribution of OM$_P$ to the higher OM$_T$. At the high NO$_x$ level, organonitrates and PAN are produced via the reaction of a peroxy radical (RO$_2$) with NO. Additionally, at the high NO$_x$ level, atmospheric processes yield fewer highly volatile chemical species (i.e., carbonyls, hydroxy carbonyls) (Hallquist et al., 2009) because OH radicals tend to react with NO$_2$ to form HNO$_3$.

A significant enhancement of gasoline SOA mass was observed in the presence of acidic seed (Fig. 3(b) vs. 3(c)) or wet seed (Fig. 3(e)). The UNIPAR-CB6r3 model simulation shows the importance of aqueous reactions of reactive organic products to increase SOA mass. In the current model, produced gasoline SOA mass is not subtracted from the gas phase. Thus, the model can cause inaccuracy particularly for a high yield SOA system. For example, the gasoline SOA yield in the presence of acidic seed under the low NO$_x$ level is high, and this presumably would cause overestimates SOA mass as seen in Fig. 3(d).

The gasoline SOA was produced in the presence of different amounts of wAS (excluding water mass) as 50 $\mu$g m$^{-3}$ (Fig. 3(e)) and 120 $\mu$g m$^{-3}$ (Fig. 3(f)). However, the influence of the concentration of wAS is trivial in both simulation and observations. This tendency indicates that a certain quantity of wAS is enough to rapidly progress aqueous reactions of reactive organic species. Figure 3(g) demonstrates the impact of the aerosol phase on SOA formation, showing a significantly different SOA growth rate after 1 PM where RH dropped lower than ERH (changing from wAS to dAS).

Figure 4 shows the impact of GWP on chamber-generated SOA (Table 1). Similar to aromatic SOA, the impact of GWP on SOA is greater with the higher NO$_x$ level because PAN and organonitrates, which are little reactive in aerosol phase, can attribute to SOA mass (Fig. 4(a)vs. Fig. 4(b)). In the presence of acidic seed (Fig. 4(c)) or wAS (Fig. 4(d)), gasoline SOA formation is less affected by GWP.

The UNIPAR-CB6r3 model simulation (Fig. 3(a) and 3(b)) in the presence of GWP was compared to that predicted with CMAQ-AE7 as seen in Fig. S7. For SOA formation via gas–particle partitioning of semivolatile compounds in CMAQ-AE7, 4 surrogate compounds produced from anthropogenic precursors (i.e., aromatic HCs, long chain alkanes, and polycyclic aromatic hydrocarbons) are employed (Qin et al., 2021). The non-volatile species that are predominantly present in particle phase are also included. The first order oligomerization reaction of organic species is included in gas mechanisms. A





distinguishable difference between two models is oligomeric fraction in SOA mass. A small fraction of SOA mass predicted with CMAQ-AE7 (Fig. S7) is attributed to oligomeric matter (~10%), while SOA mass predicted with UNIPAR-CB6r3 (Fig. 3) is dominantly attributed to $OM_{AR}$ (~80%). Laboratory studies report that 25% to 70% of SOA mass are oligomeric matter (Hallquist et al., 2009;Hall IV and Johnston, 2011;Kalberer et al., 2006). For the ozonolysis of α-pinene, Hall and Johnston

reported that the weight percentage of oligomers was estimated to be more than 50% (Hall IV and Johnston, 2011). The oligomeric fraction in 1,3,5-trimethylbenzene SOA was reported as 50-60% after 5-6 hr and increased up to 80% after 25 hr (Kalberer et al., 2006).

### 4.3 Sensitivity and Uncertainties

Figure 5 simulates the sensitivity of gasoline SOA to environmental variables (temperature, initial gasoline vapor concentration,

RH, aerosol acidity, and seed concentration). The SOA mass that predicted different environmental conditions were compared to that of the predicted at the given reference condition (60% RH, 298 K, and HC/NO$_x$=3 ppbC/ppb) under the sunlight intensity measured on 6/19/2015 (Fig. S6). The wet inorganic seed concentration was 10 μg m$^{-3}$, and the preexisting organic matter concentration was 3 μg m$^{-3}$. The initial gasoline vapor concentration was 1500 ppbC. In addition to the UNIPAR-CB6r3 simulation, the SOA formation was also predicted by the aerosol module in CMAQ-AE7.

In Fig. 5(a), the sensitivity of SOA formation to temperature was simulated between 278K and 318K. Gas–particle partitioning of reactive organic products increases with decreasing temperature, and consequently SOA formation via aqueous reaction increases. There, the high sensitivity of SOA growth to temperature appears with wet seed aerosol compared to no seeded SOA. The SOA simulation with the CMAQ-AE7 module shows a little sensitivity to temperature compared to the UNIPAR-CB6r3 simulation. Gasoline SOA formation was simulated with various initial concentrations of gasoline vapor ranging from

50 ppbC to 1500 ppbC. The SOA mass was plotted to the consumption of aromatic HCs at a given initial gasoline vapor concentration in Fig. 5(b). The consumption of aromatic HC is not linearly related to the initial gasoline vapor concentration. The SOA mass shows a curved shape indicating the partitioning contribution on the SOA growth. At the maximum SOA yield, the constant slope appears. Figure 5(c) illustrates the impact of NO$_x$ levels on gasoline SOA formation. All three simulated SOA mass are negatively correlated to NO$_x$ levels as seen in several studies (Zhou et al., 2019;Im et al., 2014). The SOA

formation in the presence of aqueous salted solution gradually increases with higher HC/NO$_x$ ratios (low NO$_x$ levels). In the absence of seed, SOA mass increases with a shape of a sigmoidal curve.

The impact of RH on gasoline SOA formation was simulated using UNIPAR-CB6r3 for different seed conditions under the assumption of LLPS between the organic and the inorganic phases (Fig. 5(d)). In current, there is no process to predict aromatic SOA formation in salted solution in CMAQ-AE7. In the presence of wAS, a relatively large decrease in SOA yields appeared

when the inorganic aerosol was effloresced at efflorescence RH (ERH, 37% of ammonium sulfate (Tang and Munkelwitz, 1994)). Within our simulation ranges (RH > 30%), AHS has no phase transition. As reported in previous work (Zhou et al., 2019), SOA formation in the LLPS mode has little sensitivity to RH above ERH.

As seen in Fig. 5(e), aerosol acidity accelerates SOA growth via acid-catalyzed reactions of organic products (Jang et al., 2002). The efficiency of aerosol acidity on SOA growth differs in NO$_x$ levels and HC ppb/seed mass. The impact of aerosol acidity

is limited by the amount of reactive organic products. At our simulation condition, gasoline SOA growth shows a plateau (at 80% of maximum SOA mass) as seen in Fig. 5(e), and weak acidity is enough to lead to the maximum acidity effect. The effect of the quantity of wet seed (wAS and wAHS) on SOA formation is represented in Fig. 5(f). SOA mass drastically increases with increasing wAS when the wAS mass concentration is less than 20 μg m$^{-3}$. In a similar manner to aerosol acidity, SOA formation reaches to a plateau because heterogeneous reactions are limited to the quantity of reactive organic compounds.

In the presence of AHS, SOA formation more quickly reaches to a plateau (5 μg m$^{-3}$ of AHS) than that with wAS.



Figure 6 represents the uncertainties of the SOA prediction when changing major model parameters ($p_{L,i}^{\circ}$, $\gamma_{in,i}$, $k_{o,i}$, and $k_{AC,i}$) in the absence of GWP. At given conditions, $\gamma_{w,i}$ and $p_{L,i}^{\circ}$ are more significantly influential on SOA prediction than $k_{o,i}$, and $k_{AC,i}$. The uncertainty in $p_{L,i}^{\circ}$ was reported as 45% based on the group contributions. The gasoline SOA mass was influenced from -25 to 50 % by increasing/decreasing $p_{L,i}^{\circ}$ as a factor of 1.5/0.5. The uncertainty associated with $\gamma_{in,i}$ ranged from -28% to 18% by increasing/decreasing as a factor of 2/0.5. The estimated uncertainties of gasoline SOA from $k_{o,i}$ and $k_{AC,i}$ are relatively as small as -13 to 9 % by increasing/decreasing them as a factor of 1.5/0.5.

## 5 Atmospheric Implication

In this study, the gasoline SOA formation potential was simulated by using the UNIPAR-CB6r3 model under varying $NO_x$ levels and seed conditions. UNIPAR-CB6r3 has a mechanism to predict SOA formation via CB6r3 gas mechanisms, gas–particle partitioning, and heterogeneous reaction in the absence of GWP bias. Through the model simulation, we conclude that both heterogeneous reactions in salted aqueous phase and the implementation of model parameters corrected for GWP are critical to accurately predict SOA mass. For example, SOA mass in Fig. 4 increases up to 1.8 times by using the corrected model parameter on account of GWP bias. As seen in Fig. 5(e) and 5(f) (model sensitivity to seed conditions), SOA mass increases by three times from including 10 µg m$^{-3}$ of wAS at 60% RH, in comparison to no-seeded SOA mass. The concentrations of $SO_2$ and sulfate have been rapidly declined due to innovative technologies and governmental efforts (Aas et al., 2019). However, the impact of the salted aqueous phase on SOA growth can be achieved even with a relatively small quantity of inorganic seed. For example, SOA formation from the photooxidation of 1500 ppbC of gasoline vapor can reach a plateau of 3-4 times higher organic mass with 5 µg m$^{-3}$ of wAHS or 20 µg m$^{-3}$ of wAS (Fig. 5(f)) under the high $NO_x$ level.

Electrolytic inorganic salts are ubiquitous in an urban ambient, because sulfate and nitrate are produced by the atmospheric oxidation of anthropogenic $SO_2$ and $NO_x$, respectively (Finlayson-Pitts and Pitts Jr, 1999). However, most current SOA modules are capable to simulate SOA formation via aqueous reactions only for a few reactive organic species (i.e., epoxydiol, glyoxal, and methylglyoxal). Numerous chemical species originating from the atmospheric oxidation of various precursors can be involved in aqueous reactions to form SOA. For example, the glyoxal fraction of the simulated gasoline SOA mass in Fig. 3(e) in the presence of wAS was about 40%, the methylglyoxal fraction was about 1%, and $OM_P$ was 26%. The remaining 23% associated with $OM_{AR}$ indicates a significant contribution of heterogeneous reactions of the reactive organic species other than glyoxal and methylglyoxal.

The liquid water content of $PM_{2.5}$ is high during regionally (i.e., eastern US) and seasonally (i.e., summer) humid conditions (Carlton et al., 2020). High humidity can shift the partitioning of hydrophilic organic gases toward the coexisting inorganic salted solution than to the organic phase, enhancing SOA growth via aqueous-reactions (Carlton and Turpin, 2013). When inorganic salt reaches the deliquescence relative humidity (DRH, 80% for AS (Brooks et al., 2002)), reactive organic species undergo aqueous phase reactions to form SOA. During warm periods at nighttime, inorganic salts can often be deliquesced in the eastern US and remain in aqueous solution above ERH. In arid areas where inorganic salts can be effloresced, SOA formation is depressed. Evidently, many field studies and model simulations have shown the greater amounts of SOA formation in eastern US in the summertime (Malm et al., 2017;Kelly et al., 2018), suggesting the important role of aqueous reactions on SOA formation.

In this study, the aerosol phase state is assumed to be LLPS for gasoline SOA formation in the presence of inorganic seed, because of hydrophobicity of some gasoline oxygenated products. The inorganic/organic mixed aerosol in the urban atmosphere, where automobile exhaust emissions and industrial solvents are abundant, may be governed in LLPS. Unlike SOA originating aromatics and terpenes, isoprene SOA is very polar and possibly mix with electrolytic aqueous solution to form a



homogeneously mixed phase under the high humidity (Beardsley and Jang, 2016;Bertram et al., 2011). However, isoprene is not the only precursory HC to form SOA. For example, terpene, a biogenic hydrocarbon, can coexist and form much less polar products than isoprene. The determination of the aerosol phase state is still controversial due to complex chemistry and precursors. LLPS likely appears in most urban areas and the regions that are influenced by high terpene emissions. In general, the atmospheric aging process increases the polarity and oligomeric matter in aerosol. Increased polarity increases organic

solubility in the salted aqueous phase, but oligomer is less favorable than low MW organics for mixing with salted aqueous phase. This is because oligomers require a large combinational energy for mixing with small water molecules (Barton, 2017).

The organic products consumed to form SOA can influence gas mechanisms, although their quantity is little. Highly reactive organic species (i.e., glyoxal and multifunctional products in group VF and F) for heterogeneous chemistry to form SOA can also be quickly photolyzed in the gas phase. Thus, the consumed organics to form SOA can possibly influence concentrations

of atmospheric oxidants and radicals. However, most SOA models including UNIPAR-CB6r3 are not capable of connecting carbon depletion to gas mechanisms due to SOA formation. This issue would potentially introduce inaccuracy in gas oxidation and aerosol prediction. Though the explicit approach (including detail gas oxidation mechanisms, partitioning, and in-particle chemistry formation) is complex and time demanding, it improves a mass balance of organic products in multiphase reactions. In UNIPAR-CB6r3, the mathematical equations associated with stoichiometric coefficients of lumping species and their

physicochemical parameters are inherited from explicit gas mechanisms. Thus, the model uncertainty can be caused by missing mechanisms in the gas phase. Additionally, the unidentified aerosol chemistry causes inaccuracy in SOA prediction and interpretation of aerosol formation mechanisms. For example, a recent study showed that the autoxidation of oxidized products can yield highly oxidized matter and increase SOA mass for certain precursors (i.e., terpenes) (Bianchi et al., 2019;Pye et al., 2019). Cross-reactions between $RO_2$ radicals can form accretion products (ROOR') with a low vapor pressure (Berndt et al.,

365 2018).

*Author contribution.* MJ designed the experiments and MJ, SH carried them out. SH prepared the manuscript with contributions from MJ.

*Competing interest.* The authors declare that they have no conflict of interest.

*Acknowledgments.* This research was supported by awards from the National Science Foundation (NSF) (AGS1923651) and the FRIEND (Fine Particle Research Initiative in East Asia Considering National Differences) Project through the National Research Foundation of Korea (NRF), funded by the Ministry of Science and ICT (2020M3G1A1114562).




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





Table 1. Experimental conditions for the photooxidation of the gasoline in the UF-APHOR chamber.

| Date (Chamber ID) | Initial condition | | | | | Temp (K) | %RH | max OM ($\mu g\ m^{-3}$) | Figure |
| | HC[a] (ppbC) | HC/NO$_x$ (ppbC/ppb) | Seed[b] | Seed mass[c] ($\mu g\ m^{-3}$) | OM$_0$[d] ($\mu g\ m^{-3}$) | | | | |
|---|---|---|---|---|---|---|---|---|---|
| 12/5/2020 (E) | 1800 | 12.8 | NS | - | 2 | 281-302 | 46-98 | 13.4 | 3(a), 4(a) |
| 03/06/2019 (W) | 1500 | 11.0 | SA | 30 | 2 | 290-315 | 28-91 | 9.6 | 3(d) |
| 1/16/2021 (E) | 1500 | 12.5 | dAS | 50 | 2 | 275-296 | 24-86 | 7.9 | 3(e) |
| 1/16/2021 (W) | 1500 | 12.5 | wAS | 50 | 2 | 276-296 | 60-93 | 20.07 | 3(e), 4(d) |
| 1/19/2021 (E) | 1500 | 12.2 | wAS | 120 | 3 | 274-300 | 47-88 | 22.8 | 3(f) |
| 1/4/2021 (E) | 1500 | 2 | SA | 30 | 4 | 277-300 | 26-88 | 8.9 | 3(b), 4(b) |
| 1/4/2021 (W) | 1500 | 2.3 | NS | - | 4 | 278-301 | 32-93 | 6.6 | 3(c), 4(c) |
| 1/28/2021 (E) | 1500 | 6.8 | wAS | 30 | 1.5 | 279-297 | 31-91 | 10.1 | 3(g) |

[a] Total HC concentrations in gasoline injected into the chamber. The HC concentrations were determined by using GC/FID (Fig. S1).

[b] NS, SA, wAS, and dAS indicate non-seeded, sulfuric acid seed, wet ammonium sulfate seed, and dry ammonium sulfate seed, respectively.

[c] The seed mass is determined as a dry mass, without water mass.

[d] The pre-existing organic matter (OM$_0$) is determined for the chamber air prior to the injection of inorganic seed and HC.





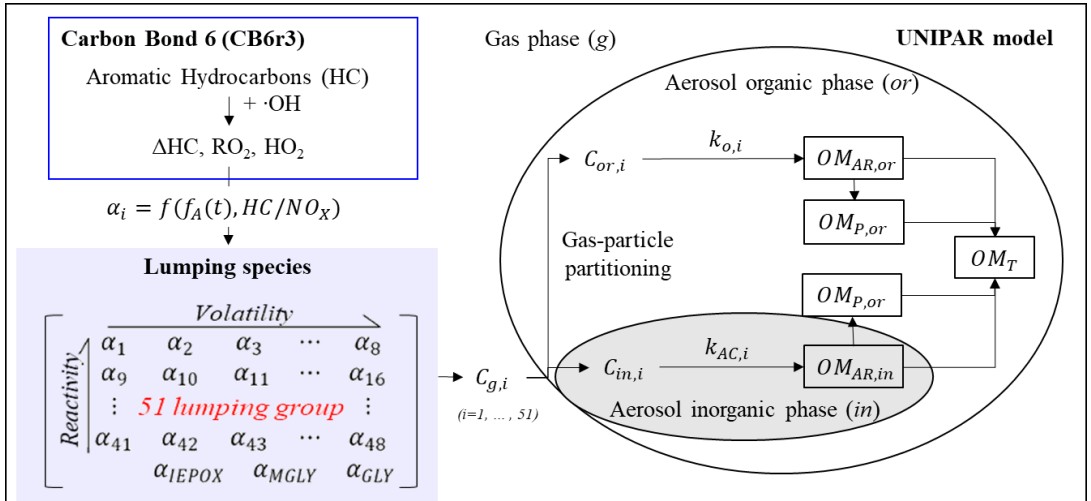

Figure 1. The structure of the UNIPAR-CB6r3 to predict the gasoline SOA formation. $C_{g,i}$, $C_{or,i}$, and $C_{in,i}$ are the concentration of organic compound ($i$) in gas phase ($g$), organic phase ($or$), and inorganic phase ($in$). The dynamic stoichiometric coefficient (dynamic $\alpha_i$), the consumption of HC ($\Delta HC$), the concentration of hydroperoxide radical ($[HO_2]$), and the concentration of organic peroxyl radical ($[RO_2]$) are simulated from the CB6r3. The aging scale factor ($f_A$) is represented as a function of $[HO_2]$, $[RO_2]$, and the initial concentration of HC (Zhou et al., 2019). $OM_{AR}$, $OM_P$, and $OM_T$ indicate organic matters (OM) formed from aerosol-phase reactions, OM formed from the partitioning process, and total OM, respectively.

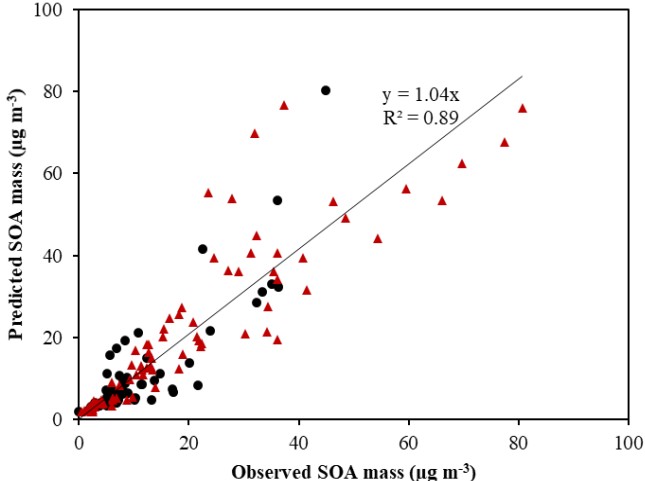

Figure 2. The linearity of predicted SOA mass ($\mu g\ m^{-3}$) using UNIPAR-CB6r3 and observed SOA mass ($\mu g\ m^{-3}$) in the absence
and the presence of wet inorganic seed. SOA mass was produced via the photooxidation of various aromatic HCs (Table S2)
in the UF-APHOR chamber.



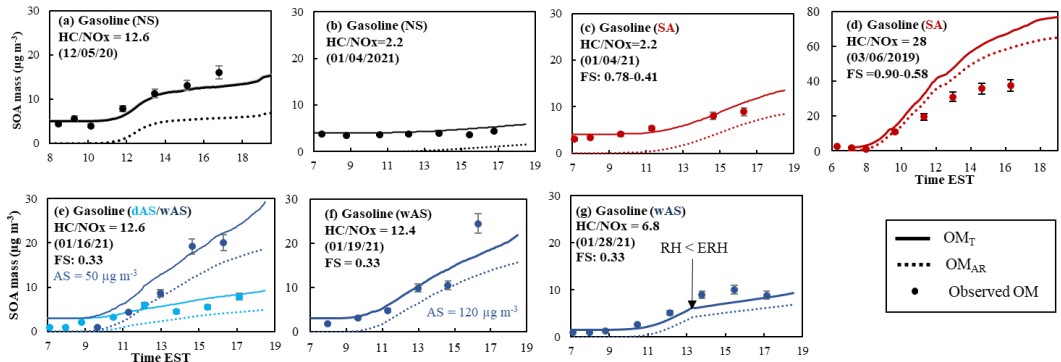

Figure 3. Observed (symbol) and simulated SOA mass using UNIPAR-CB6r3 (line) for the photooxidation of gasoline vapor
at different NOx levels. SOA mass concentrations are corrected for the particle loss to the chamber wall. The simulated $OM_T$
(solid line) and $OM_{AR}$ (dotted line) are also illustrated. The error (9%) associated with SOA mass was estimated with the
instrumental uncertainty in the OC/EC analyzer. SOA mass was produced via the photooxidation of gasoline vapor (Table 1)
in the UF-APHOR chamber.





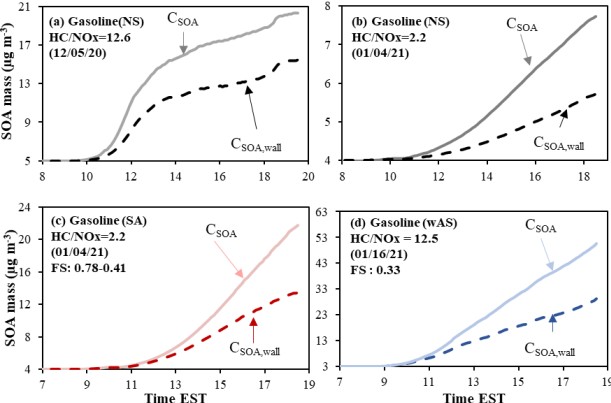


Figure 4. The comparison of gasoline SOA mass ($C_{SOA,wall}$) in the presence of GWP and gasoline SOA mass ($C_{SOA}$) in the absence of GWP. SOA formation was simulated using UNIPAR-CB6r3 for the photooxidation of gasoline vapor at given experimental conditions (Table 1).



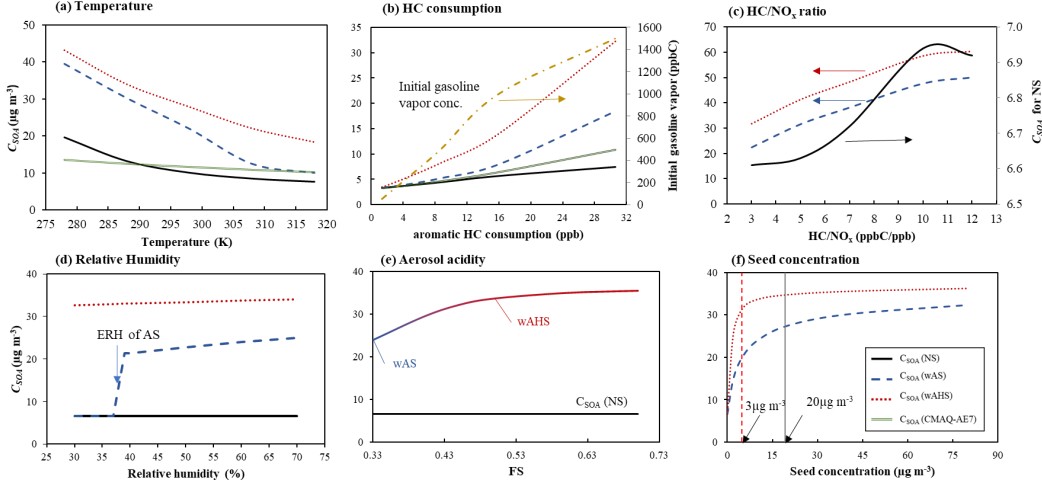


Figure 5. Sensitivities of simulated gasoline SOA mass to (a) temperature, (b) aromatic HC consumption, (c) HC/ $NO_x$ ratio, (d) relative humidity, (e) aerosol acidity, and (f) seed concentrations. SOA mass is simulated at the high $NO_x$ level (HC/$NO_x$= 3 ppbC/ppb) under given reference conditions (298K, 60% RH, and 1500 ppbC of gasoline vapor) under the specific sunlight intensity measured on 06/19/2015 (Fig. S6). The concentrations of $OM_0$ and inorganic seed are 3 μg m$^{-3}$ and 10 μg m$^{-3}$,

respectively. The simulated SOA mass using UNIPAR-CB6r3 was compared to that using CMAQ-AE7 module in Fig. 5(a) and 5(b).



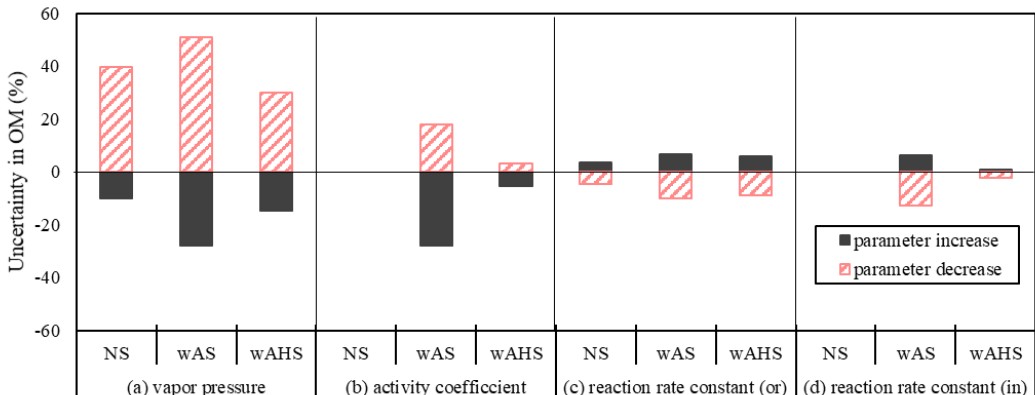

Figure 6. Uncertainties in UNIPAR-CB6r3 simulated gasoline SOA mass due to model parameters: (a) vapor pressure ($P_{L,i}$),
(b) activity coefficient in inorganic phase ($\gamma_{in}$), (c) aerosol phase reaction rate constant in organic phase ($k_o$), and (d) aqueous
phase reaction rate constant ($k_{AC}$). The errors associated with $P_{L,i}$, $k_{AC}$, and $k_o$ were estimated by increasing and decreasing
the factors by 150 and 50%. The uncertainties associated with $\gamma_{in}$ were estimated through increasing and decreasing $\gamma_{in}$ by
200 and 50%, respectively. The SOA formation was simulated at the high $NO_x$ level (HC/$NO_x$= 3 ppbC/ppb) under the given
reference conditions (298K, 60% RH, and 1500 ppbC of gasoline vapor) under the specific sunlight intensity measured on
06/19/2015 (Fig. S6). The concentration of pre-existing organic matter and inorganic seed was 3 µg m$^{-3}$ and 10 µg m$^{-3}$,
respectively.