# Peer review of "Prediction of Secondary Organic Aerosol from the Multiphase Reaction of Gasoline Vapor by Using Volatility–Reactivity Base Lumping"

_Atmospheric Chemistry and Physics, 2021_

## Author Comment (AC1)

**Response to the reviewer1 (Manuscript Ref. NO.: acp-2021-649)**

We would like to thank the reviewer for their time, and useful comments. Their comments are repeated below, followed by our response.

**General comments:**

Han and Jang provide some modeling insights on the photooxidative fates of gasoline emissions, using an SOA growth model with corrective terms that account for gas-wall partitioning phenomena that may bias kinetic inferences from experimental chamber data. Simulations across a range of $NO_x$ and seed aerosol conditions were developed and compared to observations of ambiently irradiated aerosols in the University of Florida atmospheric chamber, showing reasonable consistency between estimated and measured SOA mass. Further, the authors report broad-strokes sensitivity analyses for a variety of initial conditions and model parameters.

While the core content of the authors' work is interesting and relevant to the field at large, some minor revisions focusing on the reworking of introduction and discussion would be necessary prior to its wider release. In particular, further explanation and disambiguation of certain experimental or modeling decisions would be helpful to better reinforce the authors' assertions in their analysis. Therefore, I recommend this manuscript for publication upon the resolution of the following questions and comments.

**Specific Comments:**

1. A major takeaway of this manuscript is that it is necessary and important to implement corrections for gas-wall partitioning into SOA models. While perhaps an obvious statement to make, I believe it would be useful to underscore that GWP is a largely unavoidable artifact of the experimental data that informs SOA models and is not based in ambient atmospheric phenomena. The authors note that GWP can vary through several different operational and experimental factors; providing illustrative ranges for wall-loss rate coefficients, similar to the Introduction section of Cappa et al. 2016 (Cappa et al., 2016) will help contextualize the magnitude of these contributions to overall mass balances to the reader.

   **Response:**

   The magnitude of GWP is closely related to the wall loss rate coefficient ($kon_{w,i}$ in this paper). The range of $kon_{w,i}$ and the impact of GWP on aromatic SOA are now given in the Section 4.1. Please see L245, L248, and L253 in the revised manuscript:

   L245: "The estimated absorption rate constant ($kon_{w,i}$) of $i$ to the chamber wall was ~$5\times10^{-4}$ s$^{-1}$ for UF-APHOR chamber, differed by the volatility and functionalities of the lumping structures."
   L248: "In case of high $NO_x$ level, SOA mass from the photooxidation of benzene insignificantly increased as a factor of 1.04 after GWP correction, compared to other aromatic HCs which can produce 1.14-4.75 times higher SOA mass in the absence of GWP impact."
   L253: "The $C_{SOA}$ to $C_{SOA,wall}$ ratio is higher under the NS condition (~4.75) than that in the presence of wAHS (~2.54)."

2. Overall: Given that there are many acronyms and abbreviations used throughout this manuscript, it may be helpful (though perhaps not necessary) to include a glossary or list of abbreviations in the SI to improve general readability.

**Response:**

The list of the abbreviation and acronyms have been added in SI as a Table S3 in the revised manuscript as below:

Table S3. The list of the acronyms and abbreviation in the manuscript and their definitions.

| Acronyms or abbreviation | Definition |
|---|---|
| GWP | Gas-Wall Partitioning |
| HC | Hydrocarbon |
| QSAR | Quantitative Structure Activity Relationship |
| RH | Relative Humidity |
| OC | Organic carbon |
| $\Delta HC$ | The consumption of hydrocarbons |
| $[RO_2]$ | The concentration of $RO_2$ |
| $[HO_2]$ | The concentration of $HO_2$ |
| $\alpha_i$ | The stoichiometric coefficient of the lumping species $i$ |
| $OM_T$ | Total SOA mass |
| $OM_P$ | The SOA mass generated via gas–particle partitioning |
| $OM_{AR}$ | The SOA mass generated via heterogeneous reactions in organic and inorganic phases |
| $p_{L,i}^{\circ}$ | Vapor pressure of the lumping species $i$ (mmHg) |
| $MW_i$ | Molecular weight of the lumping species $i$ (g/mol) |
| $O{:}C_i$ | Oxygen to carbon ratio of the lumping species $i$ |
| $HB_i$ | Hydrogen bonding of the lumping species $i$ |
| $R_i$ | Reactivity scale of the lumping species $i$ in the aerosol phase |
| $C_{g,i}$ | The gas concentration of lumping species $i$ |
| $C_{or,i}$ | The concentration of lumping species $i$ partition onto the organic phase |
| $C_{in,i}$ | The concentration of lumping species $i$ partition onto the inorganic phase |
| $K_{or,i}$ | The partitioning coefficient of $i$ into the organic phase |
| $K_{in,i}$ | The partitioning coefficient of $i$ into the inorganic phase |
| $MW_{or}$ | The averaged molecular weight of $OM_T$ (g mol$^{-1}$) |
| R | The ideal gas constant (8.314 J mol$^{-1}$ K$^{-1}$) |
| T | Temperature (K) |
| $MW_{in}$ | The averaged molecular weight of inorganic aerosol (g mol$^{-1}$) |
| $\gamma_{or,i}$ | The activity coefficient of $i$ in organic phase |
| $\gamma_{in,i}$ | The activity coefficient of $i$ in inorganic phase |
| FS | Fractional sulfate |
| SA | Sulfuric acid |
| AS | Ammonium sulfate |
| $C'_{or,i}$ | The concentration of $i$ in the organic aerosol phase (mol L$^{-1}$) |
| $C'_{in,i}$ | The concentration of $i$ in the inorganic aerosol phase (mol L$^{-1}$) |
| $k_{AC,i}$ | The reaction rate constant in the aqueous phase |
| $k_{o,i}$ | The reaction rate constant in the organic phase |
| $pK_{BH_i^+}$ | The protonation equilibrium constant |
| X | The excess acidity |
| $a_w$ | The water activity |
| $[H^+]$ | The proton concentration |
| $C_{T,i}$ | The total concentration of $i$ |
| $C_{g,i}^*$ | the effective saturation concentration of $i$ |
| $OM_0$ | The pre-existing OM concentration (mol m$^{-3}$) |
| OM | The organic matter |

| | |
|---|---|
| $MW_{oli,i}$ | The molecular weight of the dimer (g mol$^{-1}$). |
| $kon_w$ | The absorption rate constant of $i$ into the chamber wall |
| $koff_w$ | The desorption rate constant of $i$ from the chamber wall |
| LLPS | Liquid-liquid phase separation |
| NS | No-seeded |
| wAHS | Wet ammonium bisulfate |
| dAS | Dry ammonium sulfate |
| $C_{SOA}$ | The aromatic SOA mass in the absence of GWP bias ($\mu$g m$^{-3}$) |
| $C_{SOA,wall}$ | The aromatic SOA mass in the presence of GWP bias ($\mu$g m$^{-3}$) |
| ERH | Efflorescence relative humidity |
| DRH | Deliquescence relative humidity |

3. Section 2 and Table 1: The information provided is likely enough to approximate or infer the duration and magnitude of sunlight that the University of Florida chamber is exposed to in each run. However, it may be helpful for the authors to provide rough estimations for the maximum actinic flux for each day so that the reader can more easily get a feel for the ranges of irradiance across experiments, much like how the ranges of temperatures and relative humidity values are presented. For instance, it is not immediately apparent that the approximate duration between dawn until dusk is 10 hours in January/December, which better justifies the experiment length mentioned on Line 84. The authors provide a reference sunlight intensity that is used in their models, though taken in the end of a Spring season instead of a Winter season. Do the authors expect differences in seasonal incident sunlight to contribute to any potential inconsistencies in results?

**Response:**
The intensity of sunlight, which is related to actinic flux changes with season and weather conditions (cloud coverage). During chamber experiments, TUVR is continuously measured (see section 2) and applied to photolysis of chemical species in gas mechanisms. The reference sunlight intensity was measured near summer solstice to cover a large aging scale. In springtime, hydrocarbon is less aged during the photooxidation of hydrocarbons than that in summer. To clarify the experimental condition, maximum sunlight intensity measured by TUVR has been added in the Table S1 in the revied manuscript.

| Date (Chamber ID) | Initial condition | | | | | Temp (K) | %RH | max OM ($\mu$g m$^{-3}$) | Max TUVR (W m$^{-2}$)[e] | Figure |
|---|---|---|---|---|---|---|---|---|---|---|
| | HC[a] (ppbC) | HC/NO$_x$ (ppbC/ppb) | Seed[b] | Seed mass[c] ($\mu$g m$^{-3}$) | OM$_0$[d] ($\mu$g m$^{-3}$) | | | | | |
| 12/5/2020 (E) | 1800 | 12.8 | NS | - | 2 | 281-302 | 46-98 | 13.4 | 19.9 | 3(a), 4(a) |
| 3/6/2019 (W) | 1500 | 11.0 | SA | 30 | 2 | 290-315 | 28-91 | 9.6 | 31.9 | 3(d) |
| 1/16/2021 (E) | 1500 | 12.5 | dAS | 50 | 2 | 275-296 | 24-86 | 7.9 | 21.3 | 3(e) |
| 1/16/2021 (W) | 1500 | 12.5 | wAS | 50 | 2 | 276-296 | 60-93 | 20.07 | 21.3 | 3(e), 4(d) |
| 1/19/2021 (E) | 1500 | 12.2 | wAS | 120 | 3 | 274-300 | 47-88 | 22.8 | 20.4 | 3(f) |
| 1/4/2021 (E) | 1500 | 2 | SA | 30 | 4 | 277-300 | 26-88 | 8.9 | 21.1 | 3(b), 4(b) |
| 1/4/2021 (W) | 1500 | 2.3 | NS | - | 4 | 278-301 | 32-93 | 6.6 | 21.1 | 3(c), 4(c) |
| 1/28/2021 (E) | 1500 | 6.8 | wAS | 30 | 1.5 | 279-297 | 31-91 | 10.1 | 23.6 | 3(g) |

The aging degree of hydrocarbon are reflected in the gas mechanism by using aging factor which is calculated from the concentration of RO$_2$ and HO$_2$ and initial injected hydrocarbon concentration and applies to the lumping species in the UNIPAR model (see section 3 in the manuscript). In addition, the TUVR data measured in spring (3/6/2021) and winter (12/5/2020) were added in Fig. S6 (Figure S5 in the revised manuscript) to show the difference in the sunlight intensities.

(a) Reference sunlight intensity (6/19/2015)   (b) Sunlight intensity during the experiment

[Figure]

[Figure]

**Figure S5**. Time profile of sunlight total ultra-violet radiation (TUVR) measured in the UF-APHOR on (a) 6/19/2015 for the reference sunlight intensity, and that on (b) 3/6/2019 and 12/5/2020 during the experiments.

4. Line 55**:** As written, it is not clear what parameter(s) the negative biases from wall losses are affecting in SOA models.

   **Response:**
   The model parameters to form SOA are typically determined on the basis of a mass balance by using chamber data. Thus, the deposition of organic vapor to the chamber wall can cause a negative bias on SOA formation. The manuscript has been revised to respond to this comment.
   "The deposition of organic vapor onto the reactor wall can cause the negative bias in SOA prediction because SOA model parameters are typically determined on the basis of a mass balance by using chamber data. (Matsunaga and Ziemann, 2010;Zhang et al., 2014;Yeh and Ziemann, 2015;Krechmer et al., 2016;Huang et al., 2018)."

5. Line 158, "FS is 1 in SA and 0.33 in AS, indicating the aerosol acidity." It is not immediately clear what the authors mean by "indicating" in this context. Is it meant that fractional sulfate can be used as a proxy for initial aerosol acidity? What ranges of FS would be expected for ambient aerosol?

   **Response:**
   Yes, in this study, fractional sulfate was used as a variable which can indicate the aerosol acidity. In the UNIPAR model, the FS value along with humidity was applied to calculate the proton concentration in aerosol and excess acidity, which are linked to the acid-catalyzed reaction of organic species in the presence of salted aqueous solution. Based on the previous field study measured by Jang et al. (2020), FS ranges from 0.334 (neutral) to 0.8 in Florida.
   L161 is updated in the revised manuscript:
   "In the model, FS, introduced to determine aerosol acidity, ranges from 0.334 for AS to 1 for SA."

6. Line 83, "before sunrise:" given that vapor wall losses are a major feature of this paper, do the authors expect that the amount of time that the initial gasoline vapor spends in the chamber prior to photoreaction will contribute to variance in yields?

   **Response:**
   The GWP process of major aromatic hydrocarbons in gasoline is negligible due to high volatility of aromatic hydrocarbons. The oxidation of aromatic hydrocarbons begins with sunrise and thus, no semivolatile oxygenated product appears before sunrise.

7. Line 84: Similar to a broader comment above, does the experimental run in March have any notably different behavior compared to the runs that took place in January/December?

**Response:**
There was a difference in meteorological conditions, such as temperature, humidity, and sunlight intensity, between the experiments due to season and cloud coverage. In general, dual chamber experiments are performed under the same meteorological condition with two different experimental conditions such as HC/NO$_x$ ratios, seed conditions, or initial HCs. For gasoline experiments, the experiment was not repeated under the same experimental condition in different seasons. However, we performed sensitivity test of SOA formation to environmental variables (temperature and humidity) in Fig. 6 (Fig.7 in the revised manuscript).

8. Section 4.1: This section refers explicitly to multiple figures and tables in the supplemental information and is difficult to interpret without having these figures open; as such, it would likely make sense that some of this information is moved into the body of the manuscript itself. Further, the first paragraph has a majority of its text describing these figures, making it difficult to parse the main assertions and conclusions that the authors are trying to articulate. This section should be reworked to improve its readability.

**Response:**
Figure S5 has been moved to the manuscript as Fig. 3 and its description was also added in the revised manuscript (Section 4.1).

[Figure]

**Figure 3.** The simulated $C_{SOA}$ and $C_{SOA,wall}$ for 10 different aromatic HCs at the given reference conditions. The SOA formation is simulated at the 298K and 60% at a given sunlight intensity (Fig. S5). The concentration of initial HC is determined to consume 100 µg m$^{-3}$ of HC at 5PM. The initial HC ppbC/NO$_x$ ppb sets to 3 and 10 for high NO$_x$ level and low NO$_x$ level, respectively. SOA masses are also obtained at 5PM. The color of the symbol indicates the seed conditions: black, blue, and red for non-seeded (NS), wet ammonium sulfate (wAS), and wet ammonium hydrogen sulfate (wAHS), respectively.

9. Section 4.2: While it is true that the majority of the observed chamber data shows agreement with the authors' OM$_T$ model, it may be helpful to include percentage errors or residuals between model and data. Potential trends in model inaccuracy across different chamber experiments and/or times-of-day would be easier to infer.

**Response:**

The averaged deviations of simulation from experimental data have been added to Fig. 3 (Fig. 4 in the revised manuscript) caption and reads now,
"The averaged deviations of simulation from experimental data are (a) 2%, (b) -14%, (c) -32%, (d) -22% and -24%, (e) -10%, and (f) 10%."

10. Line 306: When the authors refer to "uncertainty," is it correct to state that they are performing a sensitivity analysis of sorts similar to what they perform in the preceding section, though by adjusting (phenomenological) model parameters rather than environmental conditions? Do the authors expect similar sensitivity trends if GWP factors are taken into account?

**Response:**
The uncertainty in this section is to discuss the sensitivity of SOA formation to model parameters such as vapor pressure ($P_L^\circ$), activity coefficient ($\gamma_{in}$), reaction rate constant organic ($k_{o,i}$) or inorganic phase ($k_{AC,i}$). These uncertainties are tested with model parameters in the absence of GWP (GWP free). The tendency of these SOA mass uncertainties in Fig. 6 (Fig. 7 in the revised manuscript) is similar to that in the presence of GWP, previously reported in numerous studies (Zhou et al., 2019;Yu et al., 2021;Beardsley and Jang, 2016;Im et al., 2014) and reads now.
"Figure 7 represents the uncertainties of the SOA prediction caused by the uncertainties in the major model parameters ($p_{L,i}^\circ$, $\gamma_{in,i}$, $k_{o,i}$, and $k_{AC,i}$) in the absence of GWP. The tendency of these SOA mass uncertainties in the presence of GWP (Zhou et al., 2019;Yu et al., 2021;Beardsley and Jang, 2016;Im et al., 2014) was similar to those in Fig. 6 (Fig. 7 in the revised manuscript)."

**Technical Corrections:**

1. Line 38: Missing space on "bias."

   **Response:**
   The manuscript has been revised based on this comment.
   L42 in the revised manuscript: "Much effort has been given to reduce the model–measurement discrepancies by adding missing SOA precursors (McDonald et al., 2018), including heterogeneous reactions (Carlton et al., 2010), and correcting the SOA model parameters by considering gas–wall partitioning (GWP) bias (Cappa et al., 2016; Baker et al., 2015; Hayes et al., 2015)."

2. Line 82: "flam ionization" should be "flame ionization."

   **Response:**
   The manuscript has been revised based on this comment.
   L87 in the revised manuscript: "Based on the gas chromatography-flame ionization detector (GC–FID, HP-5890/Agilent Technologies 7820A) analysis of injected gasoline vapor, 30% of carbons in the gasoline were from single-ring aromatic HCs (Fig. S1)."

3. Line 350: "…but oligomer is less favorable…" should be checked for grammar.

   **Response:**
   The manuscript has been revised based on this comment.
   L366 in the revised manuscript: "Increased polarity increases organic solubility in the salted aqueous phase. However, oligomer is relatively unfavorable to be mixed with salted aqueous phase, in

comparison to the low MW organics, because oligomers require a large combinational energy for mixing with small water molecules."

4. Line 351: "This is because…" should have been checked for grammar.

**Response:**
The manuscript has been revised based on this comment.
L366 in the revised manuscript: "Increased polarity increases organic solubility in the salted aqueous phase. However, oligomer is relatively unfavorable to be mixed with salted aqueous phase, in comparison to the low MW organics, because oligomers require a large combinational energy for mixing with small water molecules."

5. Figure 2: There is no indication in text or in the caption of which set of markers corresponds to which dataset.

**Response:**
The manuscript has been revised based on this comment.

[Figure]

Figure 2. The linearity of predicted SOA mass ($\mu$g m$^{-3}$) using UNIPAR-CB6r3 and observed SOA mass ($\mu$g m$^{-3}$) in the absence and the presence of wet inorganic seed. SOA mass was produced via the photooxidation of various aromatic HCs (Table S2) in the UF-APHOR chamber.

**6.** Figure 3: What does the 9% error refer to? Instrumental resolution? Standard deviation/error across multiple samples? It would make more sense to present this error in absolute terms (i.e., in units of $\mu$g/m$^{-3}$).

**Response:**
As mentioned in the figure caption, the error (9%) was estimated with the instrumental uncertainty in the OC/EC analyzer. The error was estimated by considering particle wall loss and gas dilutions from the instrumental uncertainty.

**References:**

Cappa, C. D., Jathar, S. H., Kleeman, M. J., Docherty, K. S., Jimenez, J. L., Seinfeld, J. H., and Wexler, A. S.: Simulating secondary organic aerosol in a regional air quality model using the statistical oxidation model-Part 2: Assessing the influence of vapor wall losses, 16, 3041–3059, https://doi.org/10.5194/acp-16-3041-2016, 2016.

Beardsley, R., and Jang, M.: Simulating the SOA formation of isoprene from partitioning and aerosol phase reactions in the presence of inorganics, Atmospheric Chemistry and Physics, 16, 5993-6009, 10.5194/acp-16-5993-2016, 2016.

Im, Y., Jang, M., and Beardsley, R.: Simulation of aromatic SOA formation using the lumping model integrated with explicit gas-phase kinetic mechanisms and aerosol-phase reactions, Atmospheric Chemistry & Physics, 14, 2014.

Yu, Z., Jang, M., Zhang, T., Madhu, A., and Han, S.: Simulation of Monoterpene SOA Formation by Multiphase Reactions Using Explicit Mechanisms, ACS Earth and Space Chemistry, 2021.

Zhou, C., Jang, M., and Yu, Z.: Simulation of SOA formation from the photooxidation of monoalkylbenzenes in the presence of aqueous aerosols containing electrolytes under various NO x levels, Atmospheric Chemistry and Physics, 19, 5719-5735, 2019.

---

## Author Comment (AC2)

**Response to the reviewer 2 (Manuscript Ref. NO.: acp-2021-649)**

We would like to thank the reviewer for their time and considerate comments. Their comments are repeated below, followed by our response.

**General comments:**

The study at hand by Han and Jang deals with environmental chamber experiments at the UF-APHOR facility in Gainesville, Florida, which are interpreted using a high-level modelling framework (UNIPAR-CB6r3). The experiments look at the formation of secondary organic aerosol (SOA) from gasoline, and its hydrocarbon constituents individually. A primary focus is the importance of particle-phase chemistry in SOA formation, while a secondary focus is the treatment of gas-wall partitioning (GWP) of volatile vapors. These topics fit well within the scope of Atmospheric Chemistry and Physics.

The authors find good correlation between the model and experiment. They find that a substantial amount of gasoline SOA is due to oligomers formed in the particle phase and GWP has a significant effect on model results. Both findings are very interesting, and I believe lots can be learned from their model. However, from reading this paper alone (and not also a wide range of previous publications), it is very hard for me to deduce what the fundamental parameters were in this model simulation and which parameters were optimized / fitted for this study. This makes it very hard to gauge the significance of the conclusions. Thus, before I can recommend this paper for publication within ACP, the authors must present their approach more clearly, with an emphasis on when and how coefficients / polynomials / fits to the data are obtained (if at all) and what the model is calibrated against.

**Major Comments:**

1.  It did not become clear to me if fit parameters were used in this study to align the observed and modelled SOA mass concentrations. If I understand correctly, an important part of this modelling approach is the determination of the polynomial equation that led to all $\alpha_i$. Line 138 states "These equations are mathematically derived by using explicit products predicted from MCM v3.3.1." How is this done, as MCM does not specify an aerosol phase reactivity scale (L126)? Line 139 describes the parameters $\alpha_i$ as "dynamically predicted", what does this mean? Is there fitting to the data happening?

    **Response:**
    The formation of oxygenated products is predicted by simulating the MCM v3.3.1 gas mechanism. The resulting oxygenated products are, then, classified into 51 lumping species based on their volatility and chemical reactivity. The stoichiometric coefficients ($\alpha_i$) of 51 lumping species $i$ are determined at varying HC/NO$_x$ ratios and the different degree of gas aging, and dynamically constructed by using the mathematical equations based on the gas simulation with MCM v.3.3.1. The degree of aging is mathematically correlated with the concentration of RO$_2$ and HO$_2$, which are normalized with the initial concentration of hydrocarbons.
    To make it clear, the explanation of the creation of the dynamic $\alpha_i$ from gas mechanisms has been updated with detail description and Fig. 1 has been updated in the revised manuscript.

[Figure]

Fig. 1:

2. Figure S4 shows and it is stated throughout the manuscript that "$OM_P$", the organic mass attributed to gas-particle partitioning, is very low or even close to zero for some SOA precursors. In turn, this means that SOA mass must be almost exclusively (> 90 %) due to particle-phase reactivity ($OM_{AR}$). It is not entirely clear to me how solid the result is because I do not understand how the authors come to this conclusion. Is this just a result from the MCM-trained model, or does this rely on previous chamber experiments to optimize $\alpha_i$ parameterizations?

**Response:**
The $\alpha_i$ value was not optimized to the chamber data. As described in the response to comment 1, $\alpha_i$ is dynamically constructed by using the mathematical equations as a function of $NO_x$ level and the degree of product aging based on the gas simulation with MCMv3.3.1 (Fig. 1). The $\alpha_i$ value with the HC consumption ($\Delta$HC) enables the calculation of the concentration of lumping species. SOA forms via multiphase partitioning (gas, organic, and aqueous phases) and aerosol phase reactions (organic and aqueous phases). The $OM_P$ contribution to the total SOA depends upon precursor types, $NO_x$ levels, seed types and temperature. In general, the fraction of $OM_P$ increases with decreasing temperature, decreasing $NO_x$ levels, and decreasing aerosol acidity (Zhou et al., 2019). Additionally, aromatic SOA produces a higher $OM_{AR}$ than terpene (Yu et al., 2021).

3. Given this result, it is surprising to me that Figure 6 shows that vapor pressures are the most relevant model parameters. Is that because low vapor pressure is a prerequisite to molecules being present in the particle phase where they undergo particle-phase chemistry?

**Response:**
Lowering vapor pressure can increase the gas-particle partitioning for both reactive and non-reactive organic species for aerosol phase reactions. Thus, both $OM_P$ and $OM_{AR}$ can increase. Relatively the $OM_P$ fraction increases more than $OM_{AR}$ similarly to the impact of temperature.

4. Does $OM_{AR}$ mean that the product itself would not condense and only the particle-phase reactivity makes it stay in the particle phase? What types of reactions and products would this be that make up such a large share of SOA? I think my questions are inherently connected to point 1 above: have any of these parameters ($\alpha_i$ or $k_{o,i}$) been fitted for this study or does the model just naturally fit the data? My skepticism comes from the simulations results of gasoline, where particle-phase reactivity suddenly is of less importance (especially Figs. 3a,b,c). Why is that so?

**Response:**
$OM_{AR}$ is estimated as a second order reaction product from condensed organics based on the assumption of a self-dimerization reaction of organic compounds in media.

$\alpha_i$ is created by using the predetermined mathematical equation by using the product distribution from gas mechanism.

SOA yields are affected by the rate constant ($k_{AC,i}$) in salted aqueous reactions. In our previous work (Jang et al., 2005;Jang et al., 2006), a 2$^{nd}$ order $k_{AC,i}$ (L mol$^{-1}$ s$^{-1}$) was described via a rate determining step for polymerization to form polyacetal. The kinetic equation terms in $k_{AC,i}$ is related to both the molecular structure of an organic product (basicity of an organic compound in strong acid media and the hydration constant related to a reactivity scale); excess acidity (non-ideality of organic species in strong inorganic acid); and inorganic parameters (e.g., [H$^+$], FS, and aerosol water content). The coefficients of these model parameters in $k_{AC,i}$ were obtained via semiempirically fitting the equation to aerosol growth by heterogeneous acid-catalyzed reactions of various carbonyls with acidic inorganic aerosol using a flow reactor.

$k_{o,i}$ is determined by extrapolating $k_{AC,i}$. to the neutral condition (no acid) and in the absence of salted aqueous solution (organic phase) to process oligomerization in organic phase. Unlike aqueous reactions, $k_{o,i}$ is sensitive to the organic layer viscosity (De Schrijver and Smets, 1966;Reid et al., 2018). Thus, the additional term as a function of MW and O:C (Eq. 7) were introduced to express the sensitivity of SOA formation to organic viscosity and leveraged to numerous chamber data originating from three terpenes (α-pinene, β-pinene, and d-limonene), 10 aromatics (benzene, toluene, 3 xylene isomers, and 3 trimethylbenzene isomers), isoprene: nearly 120 data collected from the UF-APHOR large outdoor smog chamber for last 10 years) (Im et al., 2014;Beardsley and Jang, 2016;Zhou et al., 2019;Yu et al., 2021). Then, the resulting UNIPAR model and model parameters were applied to gasoline SOA of this study.

The importance of the OM$_{AR}$ or OM$_P$ is differed by meteorological and experimental conditions or the type of hydrocarbons. As mentioned as a response to the comment 2, the OM$_P$ contribution to the total SOA depends upon precursor types, NO$_x$ levels, seed types and temperature. In general, the fraction of OM$_P$ increases with decreasing temperature, decreasing NO$_x$ levels, and decreasing aerosol acidity (Zhou et al., 2019). Additionally, aromatic SOA produces a higher OM$_{AR}$ than terpene (Yu et al., 2021).

5. The reaction rate $k_{o,i}$ must be a very influential parameter, given the high fraction of OM$_{AR}$. Why is its sensitivity in Fig. 6 (Fig.7 in the revised manuscript) so small? It follows a semi-empirical description. Is there a $k_{o,i}$ for every one of the 51 lumping species for each of the 10 hydrocarbons, so 510 individual parameters? How about cross-reactions between lumping species, how are these treated?

**Response:**
$k_{o,i}$ or $k_{AC,i}$ controls reaction rates. It depends on duration of chamber simulation (10-12 hours). Within chamber simulation time scales, it is hard to see the impact of aerosol rate constant. The reaction time scale of highly reactive organic species in VF and F group are very short ranging from second to minutes.

As shown in the Eq. 7, $k_{o,i}$ is determined based on the $MW_{or}$ and $O{:}C$ of SOA, and the $pK_{BH_i^+}$ and $R_i$ of lumping species $i$. Those $pK_{BH_i^+}$ and $R_i$ of lumping species $i$ is specific for the 51 lumping species which is unified for all the aromatic hydrocarbons. Thus, we have 51 individual parameters for the oxygenated products from 10 different aromatic HCs. The cross-reaction between lumping species is not considered in this study.

6. To me, the abstract seems very technical and does not reflect the discussion in the manuscript well. For example, OM$_P$ and OM$_{AR}$ are not mentioned there.

**Response:**
The abstract has been revised based on the comment.

7. It is not clear to me what role the inorganic particle phase reactions play for this study, can the authors comment on that in the manuscript?

**Response:**
The electrolytic inorganic particle can contain a significant amount of aerosol water above ERH or DRH. When the inorganic aerosol is wet, reactive organic species can heterogeneously react in aqueous phase and increase SOA mass.
It has been added in the revised manuscript (L268).

8. Can it be calculated with the model which components of the gasoline HC mix form the most SOA?

**Response:**
No, the current UNIPAR-CB6r3 model cannot track the SOA formation form the individual aromatic hydrocarbons because 10 aromatic hydrocarbons use the unified array for physicochemical parameters. However, it is possible to modify the model to keep physicochemical parameters of individual aromatic hydrocarbons.

**Minor and Technical Comments**

1. 88 – It is not clear what CCl4 is used for in this work.

**Response:**
$CCl_4$ was injected and its concentration was monitored during the experiment to obtain the dilution factor of the chamber. The dilution factor obtained from $CCl_4$ measurement was applied to experimental data to correct the dilution during the experiment. It has been added in the revised manuscript.

2. Figure 2: Caption is missing what red and black markers are, respectively.

**Response:**
The Figure 2 has been revised based on this comment.

[Figure]

Figure 2. The linearity of predicted SOA mass ($\mu g\ m^{-3}$) using UNIPAR-CB6r3 and observed SOA mass ($\mu g\ m^{-3}$) in the absence and the presence of wet inorganic seed. SOA mass was produced via the photooxidation of various aromatic HCs (Table S2) in the UF-APHOR chamber.

3. Eq. 7 and L183: I cannot follow how with Eq.7 the impact of viscosity on oligomerization rate is considered. Why would viscosity affect chemical reaction rate?

**Response:**
The growth of organic particles can occur by in-particle chemistry; the rates of these reactions can be limited by slow bulk diffusion within a particle. It has been added in the revised manuscript L195 as:

"Studies showed that viscosity can also influence chemical reaction rate, limited by slow bulk diffusion within a particle (De Schrijver and Smets, 1966;Reid et al., 2018)."

4. Fig. S4 and L232: The model lines for benzene $OM_P$ are not visible in the figure.

**Response:**
Figure S4 has been revised based on this comment. Now, the dotted line represents the $OM_{AR}$ instead of $OM_P$ in Fig. S4.

[Figure]

**Figure S4.** Observed (plot) and simulated (line) SOA mass in the chamber studies of aromatic HCs. The simulated $OM_T$ (solid line) and $OM_{AR}$ (dotted line) are illustrated. Particle loss of experimental data onto the chamber wall was corrected. The ranges of FS are presented for experiment under the acidic condition to indicate aerosol acidity over the course of the experiment. The error (9%) associated with SOA mass was estimated with the instrumental error originating from the OC/EC analyzer.

5. Fig. 3 and S4: It was confusing to me at first that the dotted lines stand for $OM_P$ in Fig. S4 and for $OM_{AR}$ in Fig. 3. I might be good to align this.

**Response:**
The manuscript has been revised based on this comment. Now, the dotted line represents the $OM_{AR}$ instead of $OM_P$ in Fig. S4. Please see the figure in the response for the minor and technical comment 4.

**References**

Beardsley, R., and Jang, M.: Simulating the SOA formation of isoprene from partitioning and aerosol phase reactions in the presence of inorganics, Atmospheric Chemistry and Physics, 16, 5993-6009, 10.5194/acp-16-5993-2016, 2016.

De Schrijver, F., and Smets, G.: Polymerization kinetics in highly viscous media, Journal of Polymer Science Part A-1: Polymer Chemistry, 4, 2201-2210, 1966.

Im, Y., Jang, M., and Beardsley, R.: Simulation of aromatic SOA formation using the lumping model integrated with explicit gas-phase kinetic mechanisms and aerosol-phase reactions, Atmospheric Chemistry & Physics, 14, 2014.

Jang, M., Czoschke, N. M., and Northcross, A. L.: Semiempirical model for organic aerosol growth by acid-catalyzed heterogeneous reactions of organic carbonyls, Environmental science & technology, 39, 164-174, 2005.

Jang, M., Czoschke, N. M., Northcross, A. L., Cao, G., and Shaof, D.: SOA formation from partitioning and heterogeneous reactions: model study in the presence of inorganic species, Environmental science & technology, 40, 3013-3022, 2006.

Reid, J. P., Bertram, A. K., Topping, D. O., Laskin, A., Martin, S. T., Petters, M. D., Pope, F. D., and Rovelli, G.: The viscosity of atmospherically relevant organic particles, Nature communications, 9, 1-14, 2018.

Yu, Z., Jang, M., Zhang, T., Madhu, A., and Han, S.: Simulation of Monoterpene SOA Formation by Multiphase Reactions Using Explicit Mechanisms, ACS Earth and Space Chemistry, 2021.

Zhou, C., Jang, M., and Yu, Z.: Simulation of SOA formation from the photooxidation of monoalkylbenzenes in the presence of aqueous aerosols containing electrolytes under various NO x levels, Atmospheric Chemistry and Physics, 19, 5719-5735, 2019.

---

## Author Response (AR2)

**Response to Editor:**

We would like to thank the editor for the time, and useful comments. The comments are repeated below, followed by our response.

General comments:

1. Abstract: I agree with Reviewer 2 that the abstract is highly technical. There need to be stronger links to big picture questions. For example, the first sentence goes straight into what was done in the study, without mentioning any motivation or identifying any research gaps. It is not clear how the manuscript was revised in response to Reviewer 2's major comment 6.

**Response:** The abstract has been revised in the manuscript as below:

**"Abstract.** Heterogeneous chemistry of oxidized carbons in aerosol phase is known to significantly contribute to secondary organic aerosol (SOA) burdens. The UNIfied Partitioning Aerosol phase Reaction (UNIPAR) model was developed to process the multiphase chemistry of various oxygenated organics into SOA mass predictions in the presence of salted aqueous phase. In this study, the UNIPAR model simulated the SOA formation from gasoline fuel, which is a major contributor to the observed concentration of SOA in urban areas. The oxygenated products, predicted by the explicit mechanism, were lumped according to their volatility and reactivity and linked to stoichiometric coefficients which were dynamically constructed by predetermined mathematical equations at different $NO_x$ levels and degrees of gas aging. To improve the model feasibility in regional scales, the UNIPAR model was coupled with the Carbon Bond 6 (CB6r3) mechanism. CB6r3 estimated the hydrocarbon consumption and the concentration of radicals (i.e., $RO_2$ and $HO_2$) to process atmospheric aging of gas products. The organic species concentrations, estimated by stoichiometric coefficient array and the consumption of hydrocarbons, were applied to form gasoline SOA *via* multiphase partitioning and aerosol–phase reactions. To improve the gasoline SOA potential in ambient air, model parameters were also corrected for gas–wall partitioning (GWP). The simulated gasoline SOA mass was evaluated against observed data obtained in the UF-APHOR chamber under varying sunlight, $NO_x$ levels, aerosol acidity, humidity, temperature, and concentrations of aqueous salts and gasoline vapor. Overall, gasoline SOA was dominantly produced via aerosol–phase reaction, regardless of the seed conditions owing to heterogeneous reactions of reactive multifunctional organic products. Both the measured and simulated gasoline SOA was sensitive to seed conditions showing a significant increase in SOA mass with increasing aerosol acidity and water content. A considerable difference in SOA mass appeared between two inorganic aerosol states (dry aerosol vs. wet aerosol) suggesting a large difference in SOA formation potential between arid (Western United States) and humid regions (Eastern United States). Additionally, aqueous reactions of organic products increased the sensitivity of gasoline SOA formation to $NO_x$ levels as well as temperature. The impact of the chamber wall on SOA formation was generally significant, and it appeared to be higher in the absence of wet salts. Based on the evaluation of UNIPAR against chamber data from 10 aromatic hydrocarbons and gasoline fuel, we conclude that the UNIPAR model with both heterogeneous reactions and the model parameters corrected for GWP can improve the ability to accurately estimate SOA mass in regional scales."

2. Response to Reviewer 1 Specific Comment 1 seem cursory and may not be clearly addressing the comment. The response seems to provide more explanation into model details, rather than provide the big picture impacts of GWP as predicted by the model. It would be useful to expand the discussion of GWP and broader impacts, at least within the context of these experimental results. For example, why does GWP vary between aromatic hydrocarbons? Why is the factor ~1 for benzene and so much higher for other aromatic hydrocarbons?

**Response:** The section4.1 has been revised with additional discussion in the manuscript as below:

"Impact of GWP on SOA formation differs with oxidation product distributions according to volatility and reactivity. The estimated $kon_{w,i}$ of $i$ to the chamber wall was ~$5\times10^{-4}$ s$^{-1}$ for UF-APHOR chamber. Similar to $OM_P$, the impact of GWP on SOA formation is significant in the HCs with low volatile products. The HC with the higher impact of $OM_{AR}$ on $OM_T$ is less influenced by GWP. The characteristic time of aerosol–phase reactions that lead $OM_{AR}$ is generally shorter than that of GWP (Han and Jang, 2020). In the UNIPAR model, the formation of $OM_{AR}$ is irreversible to form nonvolatile oligomer products. Benzene SOA, which is mainly attributed to $OM_{AR}$, was little influenced by GWP. The oxygenated products from benzene are highly reactive in aerosol phase leading a high contribution of $OM_{AR}$ to $OM_T$, but they are volatile lowering $OM_P$ as well as the impact of GWP."

Technical comments:

Line 18: aerosol-phase reactions

**Response:** It has been updated in the revised manuscript

Line 23: United States
**Response:** It has been updated in the revised manuscript

Line 26: not sure what "in the corpus" means
**Response:** The abstract has been updated in the revised manuscript.

Line 38: "budget" is the wrong word. You do not predict a budget. The budget is the prediction itself.
**Response:** It has been updated in the revised manuscript

Line 190: "it" is incorrect, if referring to "SOA models"
**Response:** It has been updated in the revised manuscript

Line 264: peroxyacetyl nitrate is formed from RO2+NO2, not RO2+NO
**Response:** It has been updated in the revised manuscript

Line 273: overestimation of SOA mass
**Response:** It has been updated in the revised manuscript

Line 349: "in an urban atmosphere" instead of "in an urban ambient"
**Response:** It has been updated in the revised manuscript